# Daylight Saving Time Transitions: Impact on Total Mortality

**DOI:** 10.3390/ijerph17051611

**Published:** 2020-03-02

**Authors:** Michael Poteser, Hanns Moshammer

**Affiliations:** 1Department of Environmental Health, Center for Public Health, Medical University of Vienna, Kinderspitalgasse 15, 1090 Vienna, Austria; michael.poteser@meduniwien.ac.at; 2Nukus Branch of Tashkent Pediatric Medical Institute, Department of Hygiene, Nukus 230100, Uzbekistan

**Keywords:** daylight saving time, morning sunlight, total mortality, time series study

## Abstract

In Europe and many countries worldwide, a half-yearly changing time scheme has been adopted with the aim of optimizing the use of natural daylight during working hours and saving energy. Because the expected net economic benefit was not achieved, the discussion about the optimal solution has been reopened with a shifted focus on social and health related consequences. We set out to produce evidence for this discussion and analysed the impact of daylight saving time on total mortality of a general population in a time series study on daily total mortality for the years 1970–2018 in the city of Vienna, Austria. Daily deaths were modelled by Poisson regression controlling for seasonal and long-term trend, same-day and 14-day average temperature, humidity, and day of week. During the week after the spring transition a significant increase in daily total mortality of about 3% per day was observed. This was not the case during the week after the fall transition. The increase in daily mortality as observed in the week after spring DST-transition is most likely causally linked to the change in time scheme.

## 1. Introduction

In Europe and many countries worldwide, a general scheme of daylight saving time (DST) has been implemented. On a night between Saturday and Sunday in spring, clocks are set forward one hour to take advantage of the prolonged daylight with the main intention of saving energy. As the expectations for reduced net energy demand were finally not met [1,2], the European Commission decided in 2018 to discontinue DST regulation [3], opening the discussion on possible alternatives in member states. Recently, we had an inspiring coffee-table discussion with colleagues after they had published a scientific letter on that topic [4]. They expressed their preference for the implementation of a permanent standard time because of the negative impact of a lack of morning light for the circadian system and concomitant tiredness, impaired attention, and performance. However, there was a general agreement among us on the assumption that ending the bi-annual transition might be of higher relevance than the decision between summer- and standard time.

Before undertaking this study, several databases (PubMed, Cochrane, Web of Science) were searched for the term “day-light (/daylight/day light) saving time (/DST)” in combination with “mortality” without limitations. A number of studies have been retrieved that investigated the impact on daylight saving time in specific population segments or for a specific cause of death. Very few studies analyzed a general population and total mortality in this context. However, these studies are based on shorter time periods with according impact on statistical power.

Disruption of diurnal cycles could compromise the function of various organ functions, leading to increased mortality risks linked to many different causes, rendering all-cause mortality a reasonable endpoint. We further hypothesized that a part of the impact of DST transition on mortality could result from increased tiredness linked to a higher risk of accidents and professional mistakes [5].

Coincidentally, we were at that time investigating temporal changes in temperature-mortality-association [6]. For that study, we had constructed a Poisson regression model on daily all-cause mortality for the years 1970–2018 in Vienna, Austria. We decided to utilize that data to investigate the DST-related impact on total mortality.

## 2. Materials and Methods 

DST was introduced in Austria in 1980. Since then, a one hour transition was performed in spring and in fall between Saturday and Sunday: in 1980 on April 6 and September 28, in the years 1981–1995 on the last Sundays in March and September, and since 1996 on the last Sundays in March and October. 

Mortality data were obtained from the national Austrian Statistics Institute (Statistik Austria). For each death occurring in Austria since 1 January 1970, the following information was provided: Age (in years), sex, date of death, most recent place of residence (district), and primary cause of death. The latter information was provided as International Code of Diagnoses (ICD) version 8 (ICD8) until 1979, as ICD9 until 2001, and as ICD10 from 2002 onward. After 2015 data regarding the cause of death was no longer available due to personal data protection concerns. Because of the changes in diagnostic coding, the lack of information about the cause of death for the last 4 years and the lack of a hypothesis linking a specific cause of death singularly to DST, only total daily mortality was considered, among the general population of the city of Vienna.

Meteorological data (daily mean temperature and daily mean relative humidity) were abstracted from the annual reports of the Austrian Meteorological Service (Zentralanstalt für Meteorologie und Geodynamik, www.zamg.ac.at, source recording station “Hohe Warte“, western Vienna). 

In the context of DST, we expected immediate effects of a single hour of sleep lost in spring. Weekend shift workers would already be affected on Sundays, others on Mondays only. According to our colleagues’ theory [4] we would not only expect such immediate outcomes, but also more prolonged effects induced by prolonged darkness in the morning. Therefore we investigated the following days in relation to DST (scheduled at 3am on Sunday): (1) Sunday, (2) Monday, and (3) Tuesday-Friday after the spring transition, (4) Sunday, (5) Monday, and (6) Tuesday-Friday after the fall transition. We included Tuesday-Friday before the spring (7) and the fall (8) transition as negative control as well as the same 4 days as in (3), but in the 1970s (9). 

We calculated the risk ratios on the days and periods mentioned above (1–9) in a time series study using a Poisson model controlling for temporal trend, a sine-cosine function (wave-length of 365.25 days) mimicking astronomical changes in astronomical sunshine duration and thus accounting for the seasonal pattern of mortality [7], day of week, same-day relative humidity, moving average of temperature over the last 14 days, and same-day temperature (linear and quadratic term).

Because of the main hypothesis that the time-based governmental regulation of morning light would be a risk factor, controlling for natural changes in sunshine duration in the model is deemed the best approach. Astronomical variation in sunshine duration is best modelled by an annual sine-cosine function.

Statistical analyses were performed with STATA 15.1 [8].

## 3. Results

As we have reported in more detail in our previous paper [6], on average 56.4 deaths occurred per day. In spite of a growing population, the annual number of deaths declined from 1970 until about 2005 and then remained fairly stable. Daily mortality displayed a clear seasonal pattern with higher numbers in winter. Only in the last 10 or 15 years also a second peak in summer appeared that was well represented in the model by the same-day temperature. 

The Poisson regression model provided a good fit with little evidence for residual overdispersion. An alternatively fitted negative binomial regression model provided very similar estimates with an alpha of 0.0053.

Interestingly a higher mortality on Sundays at DST-transition in fall was found, followed by an equally strong protective effect on Mondays (Table 1). Being able to sleep one hour longer seems beneficial, while the higher rates on Sundays are consistent with an increased risk of accidents in people engaged in night and weekend shift work. However, the increase in deaths on that Sunday could also be an artefact simply due to the fact that this day is one hour longer. Assuming no effect on hourly mortality rate we would expect an increase in daily rates by about 4% as is indeed the case.

The loss of one hour in spring has no immediate adverse effect on total mortality, but because the Sunday in spring is one hour shorter we would indeed expect a reduction in daily mortality by about 4% (instead of only about 1%). Getting up earlier by one hour in a time with no or little morning daylight, does indeed increase mortality risk in the consecutive days (2.8% increase in daily mortality per day for Tuesday-Friday) after the introduction of the summer time in spring.

## 4. Discussion

Contrary to our own spontaneous hypothesis we did not find clear evidence of an immediate effect of the transitions on daily (or rather hourly) mortality on Sundays. We did find a rather prolonged effect with higher mortality rates in the week after the spring transition and a much weaker and shorter (Monday only) beneficial effect after the fall transition.

Our findings are in support of the hypothesis of our colleagues [4] and consistent with previously reported increased risks of myocardial infarction in the week following the spring transition [9] or even in the following two weeks [10]. That same working group also reported their own findings regarding circulatory deaths from the Veneto region in Italy. Similar to our results, they found no increase in circulatory deaths on the Mondays after the spring transition but on the following days (with a significant increase on Tuesday) [11]. In line with our findings, their meta-analysis [10] also showed a higher incidence in the week following the spring but not the fall transition. Lindenberger et al. [12] found mortality due to various causes of death increased after the spring but not the fall transition as well. Our expectation of increased risks directly after the transition [9,13,14] was only partly confirmed by our data. A cause-specific analysis of deaths would provide more detailed insight as even in the case of accidents different types display different rates of occurrence after DST transition days [15]. Yet we hope to contribute to the fierce on-going discussion [16] about optimal regulations for daylight saving time and artificial time zones.

Vienna is located rather centrally in the Middle-European time zone. A reduction in early morning sunlight might even affect more westerly regions of that time zone more severely, as was shown for traffic accidents [17] and even for cancer risks [18,19]. Even a minor sleep reduction by an average of 19 minutes could lead to severe health consequences including obesity, diabetes, cardiovascular diseases, and breast cancer [20] with relevant impacts on health-care costs. Therefore, the results from Vienna might even underestimate the true health effects of DST in other regions.

## 5. Conclusions

The results of our investigation clearly indicate that regulatory measures on time schemes do have a quantifiable impact on mortality in affected populations. This fact should be considered by policy makers and could probably provide an important argument in the on-going decision process [3] for an optimal general time scheme.

This study is the first providing evidence for general public health outcomes of daylight saving time transitions based on multiple decade observations. Using total mortality as a negative indicator for the impact on public health, we demonstrate that transition periods in daylight saving time regulations are linked to a rise in total mortality and should be avoided in the interest of general well-being.

## Figures and Tables

**Table 1 ijerph-17-01611-t001:** Risk ratios before and after daylight saving time (DST) transitions.

Day(s)	Risk Ratio (log(B))	95% Confidence Interval	*p*
Sunday after spring transition	0.991	0.948; 1.036	0.686
Monday after spring transition	0.979	0.938; 1.023	0.350
Tuesday-Friday after spring transition	***1.028***	***1.006; 1.050***	***0.012***
Sunday after fall transition	***1.049***	***1.003; 1.097***	***0.035***
Monday after fall transition	***0.941***	***0.898; 0.985***	***0.009***
Tuesday-Friday after fall transition	0.996	0.974; 1.019	0.725
Tuesday-Friday before spring transition	1.004	0.983; 1.026	0.713
Tuesday-Friday before fall transition	0.996	0.974; 1.019	0.735
Same Tuesday-Friday in spring in 1970s	0.987	0.952; 1.024	0.484

Bold and Italic: *p* < 0.05.

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
