# Peer review of "Daylight Saving Time Transitions: Impact on Total Mortality"

_ijerph, 2020, doi:10.3390/ijerph17051611_

Round 1
Reviewer 1 Report
“Saving daylight – killing people?” by Poteser and Moshammer utilises the general mortality statistics of Vienna, Austria from 1970-2018 to contribute more information to the question of what health consequences result from DST changes.
In general, this is a nice contribution (also because it is short) to the ongoing dispute of changing DST regulations, based on solid, readily available data.
However, it is really difficult to reconstruct the methods that the authors used to come to the conclusions. Although brief communications are always fun and very welcome, they still should provide the most basic ingredient of scientific publications, i.e., with the same dataset at hand, any other scientist can reconstruct the findings. This goal is not met here:
The abstract shortly mentions methods (“For that study, we had constructed a Poisson regression model”). The method section contains thoughts that would have been more appropriate at the end of the introduction (“Disruption of diurnal cycles could …”). The main methods description however is not helpful (“We calculated the risk ratios in the above mentioned Poisson model controlling for temporal trend, a sine-cosine function (wave-length of 365.25 days) mimicking astronomical changes in sunshine duration, day of week, same-day relative humidity, moving average of temperature over the last 14 days, and same-day temperature (linear and quadratic term”). While it is clear what the model was controlled for, it seems very unclear how the risk ratios were produced.
This can easily be fixed by making the description more understandable also for the broad readership that will be interested in the topic.
Important papers that pertain to the specific focus of this MS are not mentioned. The NEJM paper by Janszky et al (2008) should be incorporated (this MS contains excellent methods descriptions!). Also a recent paper by Fritz et al ("A Chronobiological Evaluation of the Acute Effects of Daylight Saving Time on Traffic Accident Risk": https://www.sciencedirect.com/science/article/pii/S0960982219316781) would be especially important because it shows that there is a gradient of the acute effects of DST changes within a time zone,. Vienna is fairly east within the CET, so it would be interesting to discuss that even higher risk ratios are to be expected for the vast majority of people who live west of Vienna in the same time zone (as far west as Galicia).
General mortality has, by itself a strong annual rhythm (e.g., see Aschoff: https://link.springer.com/chapter/10.1007/978-1-4615-6552-9_24). Has this rhythm been accounted for in the models?
When using monthly rates that make up annual rhythms, it is imperative to normalise month-length in the respective analysis. Has this study incorporated the different day length that underly daily statistics by normalising the rates to a 23 h day at the change in spring and to a 25 h day at that in autumn?
Minor points.
Title: although I like catchy titles, the effects of DST are not in any way akin to “killing”. To be more accurate and equally catchy, I suggest “Saving daylight but not lives” (very optional suggestion by the reviewer).
Line 25: March is not usually considered “early summer“, even if one incorporates global warming.
Line 33: “However, there was a general agreement …”; most of the modern assessments of DST take its chronic effects (= changing position within a time zone by moving its eastern border further east) far more seriously than the consequences of the acute transition. I would therefore be careful with such a casual statement.
Line 69: remind the readers of what “our colleagues “ refers to, by inserting reference [1] again.
Line 77: do the authors mean photoperiod when they write “sunshine duration”? Because sunshine duration often refers to cloud-cover, while photoperiod is clearly defined.
Line 97: I presume the authors meant Italics and not Italian.
Author Response
thank you for the detailed suggestions. Please find our response in the attached file!

Reviewer 2 Report
The article entitled “Saving daylight – Killing people?” is very interesting and improves knowledge on a actual debate in Western scientific community. It appears to be clearly written, and I really like the “Materials and Methods” section. In the latter section authors stated “Meteorological data (daily mean temperature and daily mean relative humidity) were abstracted from the annual reports of the Austrian Meteorological Service (Zentralanstalt für Meteorologie und Geodynamik, www.zamg.ac.at, source recording station “Hohe Warte”, western Vienna)”, however I could not read any data in the results section. Authors should report results on meteorological data and discuss them, considering previous literature.
Author Response
Thank you for your proposal.
We added a paragraph on general results (effects of seasonality and meteorological data) at the beginning of the “results” chapter, although a more detailed description is in our previous paper [2, now 5].
Reviewer 3 Report
Manuscript ID: IJERPH – 727398
Title: "Saving daylight – killing people?"
The aim of this manuscript is to test the hypothesis that saving daylight could have negative effects on health of population. To this aim, the national Austrian Statistic Institute was consulted to obtain mortality data.
The time window of observation was from 1970 to 2018. The days studied were Sunday, Monday and Tuesday-Friday in spring and fall transition.
Overall, results support the hypothesis of a negative effect on health, especially for fall transition.
The manuscript is well written and very clear. The topic is very current.
I have only a little and optional suggestion. Results, agreeing with previous observations, show a different effect between spring and fall transition. It seems that the spring effects are slower (tonic) while the fall effects are quicker (phasic). It is suggestive that in spring transition there is a delay of the clock time, while in fall transition an advance of the clock time. A comment on such difference is lacking. Could the Authors make a guess on this difference?
Author Response
Thank you for your suggestion! We discuss our finding now in more detail and in the light of additional existing literature.
Reviewer 4 Report
The introduction could use some additional references, on lines 27 and 28 about energy savings not being met by DST and the actions of the European Commission.
Minor wording changes:
line10: with the aim of optimizing
line 11: the expected economic benefit was not achieved
line 13: We set out to produce evidence for this discusssion
line 26: to take advantage of the
line 39: few studies analysed
line 97: Bold and italic
line 105: reported their own findings
Author Response
Thank you for your corrections and suggestions. Please find our point-to-point response in the attached file!
